# Signaling Pathways That Control Apoptosis in Prostate Cancer

**DOI:** 10.3390/cancers13050937

**Published:** 2021-02-24

**Authors:** Amaal Ali, George Kulik

**Affiliations:** 1Life Sciences Program, Alfaisal University, Riyadh 11533, Saudi Arabia; aaali@alfaisal.edu; 2Department of Life Sciences, Alfaisal University, Riyadh 11533, Saudi Arabia; 3Department of Cancer Biology, Wake Forest University Health Sciences, Medical Center Blvd, Winston-Salem, NC 27157, USA

**Keywords:** prostate cancer, androgen independence, CRPC, apoptosis, signaling, clinical trials

## Abstract

**Simple Summary:**

Therapies that inhibit androgen receptor signaling induce cell death by apoptosis in prostate epithelium and involution of non-malignant prostate gland, but are not effective against advanced prostate cancer. Recent evidence suggests that activation of androgen receptors in prostate stroma cells induces secretion of paracrine factors that control apoptosis in prostate epithelium. The nature of these paracrine factors, and of the signaling pathways they activate in prostate epithelial cells as well as the apoptosis regulating molecules targeted by these pathways remains unknown. This review summarizes the information on the proteins that regulate apoptosis in prostate cells, on signaling pathways that regulate these proteins, and provides an overview of clinical trials that target signaling pathways in prostate cancer. Understanding the intercellular communications and apoptosis regulation in normal prostate glands and in prostate tumors is essential for the design of curative personalized therapies for advanced prostate cancer.

**Abstract:**

Prostate cancer is the second most common malignancy and the fifth leading cancer-caused death in men worldwide. Therapies that target the androgen receptor axis induce apoptosis in normal prostates and provide temporary relief for advanced disease, yet prostate cancer that acquired androgen independence (so called castration-resistant prostate cancer, CRPC) invariably progresses to lethal disease. There is accumulating evidence that androgen receptor signaling do not regulate apoptosis and proliferation in prostate epithelial cells in a cell-autonomous fashion. Instead, androgen receptor activation in stroma compartments induces expression of unknown paracrine factors that maintain homeostasis of the prostate epithelium. This paradigm calls for new studies to identify paracrine factors and signaling pathways that control the survival of normal epithelial cells and to determine which apoptosis regulatory molecules are targeted by these pathways. This review summarizes the recent progress in understanding the mechanism of apoptosis induced by androgen ablation in prostate epithelial cells with emphasis on the roles of BCL-2 family proteins and “druggable” signaling pathways that control these proteins. A summary of the clinical trials of inhibitors of anti-apoptotic signaling pathways is also provided. Evidently, better knowledge of the apoptosis regulation in prostate epithelial cells is needed to understand mechanisms of androgen-independence and implement life-extending therapies for CRPC.

## 1. Introduction

Prostate cancer (PCa) is the second most diagnosed malignancy in men and the fifth leading cause of death from cancer worldwide. A total of 1,276,106 PCa cases and 358,989 deaths were reported across the world in 2018, making up 7.1% of all cancers diagnosed and 3.8% of all cancer-caused deaths in males [1]. On a global scale, the number of diagnosed cases of PCa is rising, although, PCa incidence varies between countries by more than 25-fold, [1,2]. The main socio-demographic factors that contribute to growing PCa incidence include increased animal fat consumption and growing elderly population [3].

Transformation of normal prostate tissue to PCa is a multistep process that involves the transition of the normal prostate to high grade prostatic intraepithelial neoplasia and then to localized and to advanced prostatic adenocarcinoma. For locally confined PCa, the choice of treatment is either surgical resection or radiation therapy or both [4,5]. However, while those therapies cure organ confined PCa, if the disease reoccurs it often metastasizes; thereby, complicating its treatment [6]. The first line therapy for locally advanced, recurrent or metastatic PCa is androgen deprivation therapy (ADT) [4,7,8,9,10,11,12,13,14]. ADT can be achieved by either surgical castration or pharmacological castration using anti-androgens such as bicalutamide or next generation therapies such as enzalutamide (MDV3100) [6,8,15,16]. Although advanced and metastatic PCa initially respond to ADT, PCa cells eventually become castration resistant causing nearly all PCa patients to relapse [4,9].

In most cases, castration resistant PCa (CRPC) cells continue to express an androgen receptor (AR), that could be aberrantly reactivated even in conditions of androgen deprivation [9]. When the next generation of AR-targeting therapies that prevent reactivation of AR entered clinical practice, a new evidence of increased proportion of CRPC that lost AR expression altogether has emerged from the analysis of resistant tumors [15,17,18].

Currently, the main approved chemotherapy for CRPC is docetaxel. Other approved treatments for CRPC include the immunotherapy agent sipuleucel-T and the taxane cabazitaxel. However, even with those drugs and many others in clinical trials, CRPC remains incurable [19]. Further complicating PCa treatment is the high degree of heterogeneity among PCa. Therefore, there is an urgent need for a novel and personalized approaches supplementing or even replacing ADT for treatment of PCa [20,21].

To find effective therapies it is essential to understand the mechanisms of androgen independence in CRPC, which is impossible without the knowledge of how androgen receptor signaling regulates the homeostasis of a normal prostate. Several forms of cell death and cell senescence may contribute to the prostate gland homeostasis, yet we choose to focus on apoptosis, since it is well documented in rodent models and is demonstrated by the analysis of human prostates that androgen deprivation induces apoptosis in the epithelial cells of the prostate gland and results in the involution of the gland [22,23].

The evidence from tissue recombination experiments and from the single cell gene expression analysis in the prostate gland suggest that AR signaling in the stroma triggers production of paracrine factors that control apoptosis in epithelial cells. The identity of these paracrine factors and of the mechanisms these factors engage are yet to be understood.

In this review we discuss recent information on apoptosis regulation in prostate epithelial cells, the role of BCL-2 family proteins in this process and the signaling pathways that target these proteins. We choose to focus on “druggable” signaling pathways for which clinically approved inhibitors have been developed and to provide an overview of clinical trials that test inhibitors of the anti-apoptotic signaling pathways in CRPC.

## 2. Apoptosis in Prostate Epithelial Cells Is Controlled by AR Signaling in Stroma

### 2.1. Stroma and Luminal Compartments of the Prostate Gland

The human prostate gland consists of two cellular compartments: an epithelial compartment that contains luminal and basal epithelial cells, and a stromal compartment which surrounds the epithelial layer. Luminal epithelial cells are polarized, columnar cells that form a layer that lines the lumen of the prostate. The luminal epithelium expresses high levels of the AR and cytokeratins 8 and 18, CD57, NKX3.1, as well as secretory proteins such as the prostate specific antigen (PSA). The basal epithelial cells are non-secretory cells that line the basement membrane and separate the lumen from the stroma. The basal epithelium expresses the cellular markers cytokeratin 5 and 14, p63, CD44, and GSTP1. Unlike luminal cells, basal cells express low AR levels. Rare neuroendocrine cells are also found within the basal compartment. Neuroendocrine cells are AR-negative cells that secrete neuropeptides, and the growth factors for luminal cells. Neuroendocrine cells also frequently display dendritic-like processes that can contact luminal cells. The stromal compartment surrounds the epithelial compartment and contains a large number of fibroblasts that secrete components of the extracellular matrix that maintains organ integrity and mediates signaling by growth factors; and also contains a layer of smooth muscle cells lining the epithelium that can contract and aid in the forceful discharge of the prostatic fluid. The stroma contains other components including blood vessels, lymphatics, nerves, and immune cells [4,24] (Figure 1).

### 2.2. Androgen Ablation Triggers Apoptosis

The normal growth, differentiation and functioning of the prostate gland as well as the growth and progression of PCa depends on signaling by the AR activated by the steroid hormones androgens [7,25,26,27]. Two androgens, testosterone and its more potent metabolite, dihydrotestosterone (DHT), can regulate the AR’s transcriptional activity [25]. In the prostate, testosterone, the primary circulating androgen in men, is converted to the more potent and major AR-bound androgen DHT, by the enzyme 5α[9]-reductase [25,26]. In the absence of androgens AR is sequestered in the cytoplasm by heat shock proteins (HSPs) and present in an inactive yet poised-for-activation state [7,25]. Upon binding of androgens, AR undergoes conformational change [28], dissociates from HSPs and is activated [7,25,26,27]. Active AR dimerize and translocate to the nucleus where they bind through DNA-binding domain to androgen-responsive element (ARE) in the promoter and enhancer regions of AR’s target genes [27]. Coactivators, complexes that modify chromatin, and RNA polymerase II are then recruited, and transcription of the target genes is induced or repressed [7].

Pioneering work of Huggins and Hodges on involution of the prostate gland after bilateral orchiectomy in dogs [29] was followed by studies by J.F. Kerr in rats that demonstrated dramatic involution of the ventral prostate gland (VP) after castration, with marked decline of organ weight on day 3 post castration that continued until days 15–21 [22]. Detailed analysis by electron microscopy (EM) showed the presence of autophagic vacuoles, condensation and fragmentation of cytoplasm and nuclei as well as the presence of fragmented cells engulfed by macrophages and neighboring epithelial cells indicative of cell death by apoptosis [22,30]. Time course analysis demonstrated an increased percentage of apoptosis on day 2 with maximal increase in percent of apoptotic cells at day 3 post castration that declined from day 4 and until days 13 and 20 [22,31]. The majority of apoptotic cells were found in the epithelial layer although occasionally clusters of apoptotic bodies were found in the acinar lumina [22]. These EM and cytohistologic data were interpreted as an evidence of cell death by apoptosis as a main cause of VP involution in castrated rats.

The analysis of mechanisms of VP involution in castrated rats was expanded by John Isaacs’ group that demonstrated DNA fragmentation in epithelial cells, a classic apoptosis hallmark. Based on a detailed analysis of the apoptotic index by TUNEL (terminal deoxynucleotidyl transferase dUTP nick end labeling), a method that detects cells with fragmented DNA and analysis of mitotic index by in-vivo ^3^H-tymidine labeling they concluded that involution of VP can be explained by the combination of decreased proliferation and increased apoptosis [23,32]. The leading role of apoptosis in prostate involution is further supported by experiments with prostate-restricted expression of BCL2. Transgenic mice that expressed BCL2 under probasin promoters showed a significantly lower percent of apoptotic cells after androgen ablation when compared to wild-type mice [33].

### 2.3. Role of Stromal AR in Epithelial Apoptosis Regulation

Studies in animal models by several research groups reported that apoptosis was found mostly in columnar epithelial cells that were eliminated after androgen deprivation, yet regenerated when androgen levels were restored [23]. The question remained, however, whether AR functions in “cell autonomous fashion” by directly regulating expression of genes that control apoptosis in luminal epithelial cells, or apoptosis is regulated by paracrine factors secreted in response to androgens.

To examine the cell autonomous role of AR in apoptosis, tissue recombination experiments were conducted to combine epithelium from *Tfm* mice (that lack AR expression due to spontaneous mutation on the X chromosome) with stroma from wild-type mice. In the recombined prostate tissue that expresses AR only in stromal cells, castration induced apoptosis in epithelial cells lacking AR with an apoptotic index nearly the same as that of wild-type mice. Administration of testosterone and dihydrotestosterone equally reversed apoptosis in AR-negative and AR-positive epithelial cells. These data suggests that apoptosis in luminal epithelial cells is not directly regulated by epithelial AR, but instead is regulated by paracrine factors (for example FGF10) induced by androgens through AR expressed in stromal cells [34,35]. At the same time, experiments in transgenic mice with AR knockout driven by Pb promoter showed increased apoptosis in CK8-positive luminal epithelial cells, increased proliferation in CK5-positive basal epithelial cells, and stromal atrophy [36,37,38]. The earlier report showed increased Ki-67 staining and increased apoptosis in prostate secretory epithelial cells in transgenic mice with overexpression of AR driven by Pb promoter [39]. These discordant reports on the role of AR in regulation of apoptosis and proliferation from the groups that used tissue recombination and Pb-driven transgenes illustrate challenges of analysis of the inter-cellular and intra-cellular communications that regulate homeostasis in prostate tissue.

Experiments with selective AR knockout in smooth muscle cells [40,41]; in stromal fibroblasts [42] and in both smooth muscle and stroma fibroblasts [43] (reviewed in [44]) point at a paracrine mechanism. Thus, activation of AR in prostate stroma cells induces expression of signaling molecules (FGFs, IGF-I, and others) that in turn control survival of luminal epithelial cells and morphogenesis of the prostate gland.

Perhaps the most conclusive evidence comes from the publication that used tamoxifen-activated CRE to induce AR knockout in basal and luminal prostate epithelial cells of 8-week-old mice and followed apoptosis, proliferation and gene expression at a single cell level. AR knockout in luminal cells did not change apoptosis, involution and regeneration of luminal cells during the castration/regeneration cycle [45].

Another recent report on the single cell RNAseq profiling of anterior prostate in mice provided a comprehensive analysis of cell populations that constitute epithelial and stromal compartments and their dynamics during the castration-regeneration cycle [35]. Luminal secretory cells that constituted the majority in the epithelial compartment were identified as the main cell type that contributes to regeneration by increased proliferation after circulating androgen levels are restored. The profiling of stroma cells demonstrated substantial changes in the expression of growth factor genes (*Nrg2, Igf1, Fgf10, Rspo3*) and increased expression of corresponding receptors FGFR2 and LGR4 in luminal epithelial cells in response to androgen. The importance of the FGF/ERK pathway for the ADT-resistance has been recently demonstrated in several PCa cell lines [17].

In addition to the induction of apoptosis in epithelial cells, apoptosis in endothelial cells of the prostate stroma has been reported [46]. Decreased blood flow [47] and hypoxia [48] have been also connected with apoptosis of prostate epithelial cells, however, later studies did not detect hypoxia after castration [28].

Altogether results from experiments with tissue recombination, prostate-restricted transgenes, and single cell RNAseq analysis suggest that despite high AR expression in prostate epithelia, the growth and survival of prostate epithelial cells is not directly regulated by epithelial AR. Instead, AR-dependent expression of growth and survival factors by cells in stromal compartment regulates proliferation and survival of luminal epithelial cells in paracrine fashion (Figure 1).

New evidence on indirect paracrine regulation of the prostate epithelial cells through AR signaling in the stroma challenges the current practice of indiscriminate androgen ablation therapy and harbingers a need for therapies selectively targeting AR’s in the tumor stroma [38]. More experiments utilizing inducible transgenes in combination with the single cell genomic and proteomic studies are needed to identify paracrine factors that control apoptosis in prostate cancer cells after androgen ablation. Equally important is to understand which apoptosis regulatory molecules are controlled by these paracrine signals.

### 2.4. Roles of Mitochondrial and Death Receptor-Induced Apoptosis in Prostate Involution 

Apoptosis in prostate epithelial cells can be induced by mitochondrial (intrinsic), or death receptor (extrinsic), pathways. The extrinsic apoptotic pathway is initiated by activation of death receptors belonging to the tumor necrosis factor (TNF) family: TNFRSF1A, FAS (CD95, or APO-1), TNFSF10/TRAIL receptors TNFRSF10A/DR4, TNFRSF10B/DR5, TNFRSF25/DR3 and TNFRSF21/DR6 [49]. When bound by their ligands, death receptors cluster to form the death-inducing signaling complex (DISC) that activates the initiator caspases 8 and 10. Active caspases 8 and 10 then activate the effector caspases 3, 6 and 7 that cleave substrates resulting in apoptosis. The extrinsic apoptotic pathway can be inhibited by the protein CFLAR/FLIP [50].

FAS and FASLG/FASL are expressed in mouse and human prostate tissues as well as PCa cell lines [51,52,53,54]. The paper by Suzuki et al. reported that prostates in *lpr* mice (lacking functional FAS) do not undergo involution in response to castration [55]. Another study showed 3–5-fold upregulation of FASL mRNA and protein observed 3 days post-castration, however it did not find significant differences between wild type and *lpr* mice in castration-induced VP regression or in the counts of apoptotic cells assessed by TUNEL and by morphology of apoptotic bodies [56]. A subsequent study that compared castration-induced prostate involution in wild type, *Tnf^-/-^*, *Tnfr1^-/^*^-^, *Trail^-/-^* or *lpr* mice demonstrated that diminished prostate involution was evident only in *Tnf^-/^*and *Tnfr1^-/^*^-^mice with impaired TNFR1 signaling, but not in *lpr* mice. Delay in prostate involution was also observed in mice injected with TNF-R2-Fc, a soluble TNF-R2 that can prevent activation of membrane bound TNFR1 by TNF-α. Conversely, increased production of TNF-α by the stroma of castrated mice was reported, whereas, injecting castrated *Tnf*^-/-^ mice with TNF-α restored prostate involution to the levels seen in WT mice. Data from this publication support the hypothesis that androgen ablation induces TNF-α production by prostate stroma that act on TNFR1 in luminal epithelial cells to induce apoptosis and prostate involution. However, despite slower prostate involution, no differences in apoptotic indices between WT and Tnf^-/-^ or Tnfr1^-/-^ mice was found. Furthermore, TNF-α did not induce involution in non-castrated mice, suggesting that castration not only increases TNF-α levels but also makes luminal cells sensitive to TNF-α induced apoptosis. A possible candidate for AR-regulated gene that increases sensitivity to death receptor induced apoptosis could be FLIP, as it acts as the DISC inhibitor downstream of death receptors. Regulation of FLIP by AR pathway was demonstrated in experiments on castrated rats and in prostate cells lines where FLIP levels defined sensitivity to apoptosis induced by TRAIL [57,58,59,60].

Evidently more follow-up studies with an inducible knockout of specific death receptors and downstream components of death receptor signaling cascades are needed to clarify the role of death receptors in prostate involution after androgen ablation.

The extrinsic and intrinsic apoptosis pathways are linked by the BH3-only protein BID, yet the role of BID in castration-induced apoptosis was not directly assessed. Instead, an analysis of a mitochondrial pathway in a post-castration prostate gland focused on BAX and BCL2 proteins. Increased levels of BAX and BCL2 were reported in prostate glands of mice after castration [61]. An analysis of apoptosis by counting the percent of TUNEL positive cells in the ventral prostate gland of mice with BAX knockout (*Bax*^-/-^) showed significantly less apoptosis on day 5 post-castration as compared to wild-type mice, instead, a higher percent of TUNEL positive cells was detected in *Bax*^-/-^ mice on day 1 and day 14 (although the difference did not reach statistical significance). It is worth noting that in other models knockout studies demonstrated redundancy of BAX and BAK [62]. Thus, double knockouts could be necessary to demonstrate the roles of BAX and BAK in castration-induced apoptosis. Similar to the results in *Bax*^-/-^ mice, a significant decrease in the apoptosis index was observed on day 5 in transgenic mice with BCL2 overexpression targeted to prostate gland by probasin promoters [33]; however, no data on the prostate involution in these mice were reported. Another group generated mice expressing *Bcl2* transgene under C3(1) promoter, yet no data on castration-induced apoptosis or prostate involution was reported either [63].

Apparently, inactivation of AR in the prostate stroma leads to the increased production of death receptor ligands and the decreased production of survival factors that induce apoptosis in epithelial cells (Figure 1). Altogether, results from transgenic mice suggest that apoptosis regulatory proteins involved in death receptor and mitochondrial pathways collectively regulate apoptosis induced by androgen ablation in prostate gland [64,65,66,67].

## 3. BCL-2 Family Proteins in Prostate Cancer

Analysis of gene expression repositories and published data identified BCL2, BCL2L1/BCLX, MCL1, BAX, BAK, BAD, BCL2L11/BIM, BBC3/PUMA, PMAIP1/NOXA, BIK and BID as members of the BCL-2 family proteins expressed in a normal prostate and in PCa. Several reviews discuss the contributions of specific anti-apoptotic BCL-2 family proteins to prostate cancer therapy resistance and the targeting of the BCL-2 family by siRNA, antisense DNA or BH3-mimetics [68,69,70]; however, a comprehensive systems analysis of BCL2 proteins in prostate cancer, and their changes in response to anti-cancer therapies and to anti-apoptotic signals has yet to be performed. Information on the BCL-2 family proteins in prostate cancer is summarized below.

### 3.1. Anti-Apoptotic Proteins

BCL2—the expression levels of BCL2 are regulated by transcription factors including p53 [71], WT1 [72], and NF-𝜅B [73] and by promoter methylation [74]. Additionally, phosphorylation of BCL2 induced by pro-apoptotic and by anti-apoptotic agents (taxol, IL-3, erythropoietin and bryostatin-1) have been reported. Recent publication demonstrated that BCL2 phosphorylation increased interaction of BCL2 with BAX and BAK; yet decreased interaction between BCL2 and BH3-only pro-apoptotic proteins BIM, PUMA and BAD, and suggested that the ultimate effect of BCL2 phosphorylation on apoptosis is determined by the relative abundance of these pro-apoptotic proteins [75].

In the mid to late 90s several groups assessed the roles of BCL2 protein in apoptosis induced in prostate epithelial cells by androgen ablation [76,77]. Increased BCL2 expression after castration was reported in secretory epithelial cells and in basal cells [61,78]. Decreased BCL2 mRNA levels and increased BCL2 promoter methylation was reported in prostate cancer compared to adjacent normal tissue [74]. On the other hand, analysis of changes in expression of androgen regulated genes in human benign prostate tissues xenografts revealed upregulation of BCL2 in response to androgen withdrawal. No changes in the expression of other genes involved in apoptosis regulation has been detected [79]. Earlier immunohistochemical studies reported increased levels of BCL2 protein in prostates of castrated patients [80] and in advanced prostate cancer [78,81]. A negative correlation between AR signaling and BCL2 expression was reported in LAPC4, LAPC9 and LNCaP cells [82] and in prostate tumors [83]. Apparently, increased expression of BCL2 represents a compensatory response by prostate epithelial cells to resist apoptosis induced by androgen ablation and acquire androgen-independence. Thus, ectopic BCL2 expression decreased apoptosis in prostate cancer cells and facilitated the transitioning of PCa cells to androgen-independence in vitro and in vivo [33,56,84]. Still, endogenous BCL2 is likely playing a secondary role compared to BCLX and MCL1 that expressed at higher levels in prostate epithelial cells [85].

BCLX—BCLX mRNA expression can be controlled by the transcription factors STAT [86], Rel/NF-𝜅B and ETS [87]; whereas, splicing of the BCLX gene provides another level of BCLX regulation [88]. Post-translational modifications of BCLX by phosphorylation [89] caspase cleavage [90] and deamidation [91] have been reported.

Expression of BCLX was assessed in normal prostate cells, prostate tumors and prostate cancer cell lines [92,93,94,95]. Increased BCLX mRNA and protein levels were detected in higher grade tumors, in lymph node metastases and in distant metastases [92,93,96]. A positive correlation between BCLX expression and androgen independence was also reported [93]. At the same time, in benign prostate hyperplasia a decreased level of BCLX was found and treatment with 5α-reductase inhibitor finasteride did not change BCLX expression [97].

Experiments in prostate cancer cells demonstrated binding of AR to BCLX promoter and AR-dependent regulation of BCLX expression [96]. However recent report on indirect activation of BCLX promoter via AR-> integrin α6->NF*k*B mechanism provided more nuanced interpretation of androgen-dependency of BCLX expression in prostate cancer cells [98].

MCL1 mRNA expression analysis suggests that it is the dominant anti-apoptotic protein of the BCL-2 family in prostate cancer cells. Compared to other anti-apoptotic proteins, MCL1 has a longer N-terminus and exists in two forms: a full-length protein localized at the outer mitochondrial membrane that is involved in apoptosis regulation, and a truncated protein that is localized in the mitochondrial matrix and is involved in the mitochondrial dynamics and metabolism. MCL1 is distinguished from other anti-apoptotic members of BCL-2 family by a relatively short half-life, which provides an opportunity for the dynamic regulation of MCL1 protein levels by signaling pathways that control proteostasis.

The activity and levels of MCL1 can be modulated transcriptionally, post-transcriptionally and post-translationally. Transcription of MCL1 can be changed in response to growth factors, cytokines, endoplasmic reticulum (ER) stress, hypoxia and microtubule disruption. Transcription of MCL1 can be regulated by a wide range of transcription factors such as HIF-1, SRF, CREB, c-MYC, ATF5 [99,100]. At the post-transcriptional level, MCL1 can be regulated via pre-mRNA splicing and regulation of the turnover of its short-lived mRNA (half-life of around 2–3 h) by the RNA-binding proteins and the multiple regulatory RNAs. Ubiquitination of MCL1 results in its proteasomal degradation, whereas deubiquitination halts MCL1 degradation [100,101,102]. MCL1 can be phosphorylated on numerous sites and depending on a specific site(s) it can increase or decrease anti-apoptotic capacity. Phosphorylation can modulate interactions between MCL1 and pro-apoptotic proteins, or it can regulate MCL1 protein stability through interactions with ubiquitin ligases and deubiquitinases [100].

Increased MCL1 expression was reported in PCa, in bone metastases [92,103], and in stem/tumor-initiating cells isolated from prostate tumors [104]. Upregulated MCL1 expression was found in androgen deprived PCa cells both in vitro and in vivo. AR signaling indirectly suppresses mRNA and protein levels of MCL1, but AR does not bind the MCL1 gene and the regulatory mechanism is not understood [104]. Increased MCL1 protein levels in LNCaP cells were connected with autocrine IL6 signaling; yet, no mechanistic data were provided [105]. Activation of ADRB2/PKA signaling protected prostate cancer cells from apoptosis by increasing MCL1 expression via a transcription-independent mechanism [106]. Active FLT/VEGFR1, FGFR1 and PDGFR family tyrosine kinases increased the transcription of MCL1 mRNA [103,107] in PCa cell lines. Conversely, inhibitors of tyrosine kinases increased the degradation of MCL1 [108,109]. The agents that induce ER stress or inhibit protein synthesis decreased MCL1 expression and sensitized prostate cancer cells to apoptosis [95,110].

Compared to other anti-apoptotic proteins, MCL1 stands out as a convergence node of several signaling pathways that dynamically regulate MCL1 expression levels.

### 3.2. Pro-Apoptotic Effector Proteins

BAX and BAK—can oligomerize and form pores in the outer mitochondrial membrane. BAK is constitutively localized at the mitochondria and undergoes conformational changes during apoptosis that permit homo-oligomerization [65]. BAX is present in a cytosol of healthy cells, and translocates to the mitochondria during apoptosis. Transcription of BAX and BAK can be regulated epigenetically by methylation which suppresses their expression [111].

An analysis of BAX in castrated mouse prostates showed an increased expression in secretory epithelial cells; however, experiments in mice with global knockout of BAX did not show significant differences in the apoptosis index compared to wild type mice [33]. An analysis of prostates from patients treated with radiation therapy showed an increased expression of BAX in prostate intraepithelial neoplasia versus normal glands in the same patient [112], and an increased expression of BAX was associated with poor outcomes [113], yet a mechanistic basis of increased BAX expression is unclear and the role of BAX in prostate cancer remains open.

An analysis of BAK expression by Western blotting showed no signal in prostate tissue extracts, yet immunohistochemical analysis identified BAK expression in the smooth muscle and the basal epithelial cells of prostate glands [114]. Expression of BAK was detected in prostate cancer cell lines and was increased in by estramustine, beta-lactone and other experimental therapeutics; yet, the mechanism(s) of increased BAK expression were not assessed [115,116,117]. Negative regulation of BAK by MAPK7/BMK1 phosphorylation at Y108 has been reported in several cancer cell lines [118]. BMK1 expression is increased in CRPC, however, since many receptor tyrosine kinases are activated by BMK1 in prostate cancer [119], the relative importance of BAK phosphorylation for CRPC compared to other BMK substrates is difficult to evaluate. Perhaps conditional knock-out experiments that delete BAK prior to castration, or knock-in experiments that replace WT-BAK with Y108-deficient mutants may clarify the role of BAK in apoptosis induced by androgen ablation and the role of BAK phosphorylation in apoptosis resistance in advanced prostate cancer.

### 3.3. BH3-Only Proteins

BAD—Activity of BAD can be post-translationally controlled by phosphorylation that prevents interactions with the anti-apoptotic BCL-2 family proteins. Dephosphorylation by serine/threonine phosphatases restores BAD apoptotic activity [120,121,122]. Additionally, BAD expression could be silenced epigenetically via promoter methylation [111]. In a non-phosphorylated state BAD binds to and inactivates BCLX and BCL2, but cannot bind to MCL1. Design of the first-in-class BH3 mimetic drug ABT737 is mimicking the BH3 domain of BAD [123].

Increased expression of BAD was reported in prostate carcinoma and correlated with a longer time to biochemical relapse and overall survival [124,125].

Multiple signaling pathways induced by EGF, G-protein coupled receptors (GPCR) agonists, PMA, Galectin3, omega3- fatty acids, clusterin and α2-macroglobulin as well as by the loss of PTEN or hyperactive protein kinases have been connected to BAD phosphorylation in prostate cancer [126,127,128,129,130,131,132,133,134]. Numerous protein kinases including AKT1, MAPK1/ERK, RPS6KA1/RSK1, cAMP-activated protein kinase (PKA), PAK1, PRKCI/nPKC-iota, PRKCE/nPKC-epsilon and PIM1 were reported to phosphorylate BAD in prostate cancer cells [135,136,137].

Phosphorylations at S75 and S99 (that correspond to S112 and S136 in mouse protein) create binding sites for 14-3-3 chaperons that sequester BAD in cytoplasm and prevent interactions with BCLX or BCL2 at mitochondria [11,138,139]. Phosphorylation at S118 (S155 in mice) in BH3 domain directly disrupts interaction with anti-apoptotic proteins.

Beside these three major phosphorylation sites, phosphorylation of S111 and S134 were also reported; however, their significance for apoptosis in prostate cancer cells has not been assessed [140].

BIM—is a BH3 only protein that can bind all anti-apoptotic proteins of the BCL-2 family and it also has a dynein binding motif that mediates binding to the cytoskeleton. A comprehensive review on BIM has been recently published [141]. BIM can be regulated both transcriptionally and post-translationally. At the transcriptional level, BIM mRNA levels are positively regulated by the transcription factors FOXO3, CEBPA, DDIT3/CHOP, and E2F1 and negatively regulated by the miRNA cluster miRNA-17-92. The opposing effects of cytokines and oxidative stress on BIM mRNA stability via heat-shock proteins have been reported [141]. Several forms of BIM are generated by alternative splicing including BIM-gamma expressed in prostate cancer cells [142]. At the post-translational level, BIM is regulated by phosphorylation, which have opposing effects on pro-apoptotic BIM function depending on the kinases involved and the sites of phosphorylation. Anti-apoptotic signals by MEK1/ERK/RSK1, PI3K/AKT1 and LYN kinases induce BIM phosphorylation that disrupts interactions with anti-apoptotic proteins and creates binding sites for ubiquitin ligases that stimulate its ubiquitination and subsequent proteasomal degradation [120,143,144]. In contrast, phosphorylation by PKA, MAPK8/JNK and MAPK14/p38 increases BIM apoptotic activity [141].

Elevated expression of BIM in prostate tumors has been reported, [145]. BIM expression has been detected in prostate cancer cells, and modulation of BIM levels by cytotoxic and survival signals was connected with apoptosis in prostate cancer cells and in the prostate glands of *Pten^-/-^* mice [146,147,148]. At the same time, a more nuanced role of BIM in apoptosis was proposed, when increased BIM expression had an anti-apoptotic effect that was reversed by phosphorylation that prompted interaction with BCLX and MCL1 in prostate cancer cells [145].

PUMA and NOXA—PUMA and NOXA are regulated at the transcriptional level by TP53 as well as by E2F1 [120,149]. Furthermore, transcription of NOXA can be epigenetically silenced via promoter methylation [150]. Unlike other BH3-only proteins, PUMA is only regulated transcriptionally [149]. NOXA, on the other hand, is also regulated post-translationally by ubiquitination which targets it for proteasomal degradation, and by phosphorylation at S13 position which blocks NOXA’s pro-apoptotic activity [151].

PUMA and NOXA show reciprocal expression in prostate cancer cell lines. The immunohistochemical analysis of 51 normal prostates, 64 primary PCa and 30 CRPC showed significantly less NOXA expression in normal prostates when compared to hormone resistant and hormone sensitive cancers (that showed the highest expression). Patients with a higher NOXA levels had a shorter progression-free survival, and the opposite trend was noted for PUMA [152]. In another study PUMA was detected in six BPH samples but was undetectable in five Gleason grade 4, 5 prostate cancer samples. These contradictory reports call for additional studies with more stringent controls for the specificities of antibodies. In prostate cancer cells lines and in mouse prostates the activation of ER-beta induced PUMA expression and apoptosis via FOXO3 [153]. Other studies confirmed the role of increased expression of PUMA and NOXA in apoptosis induced by cytotoxic therapies in prostate cancer cells [154], yet more in depth studies are needed to reconcile tissue culture experiments and studies of clinical samples.

BIK—is a BH3-only protein that preferentially interacts with BCLX, BCL2L2/BCL-W and BCL2A1. At the transcriptional level, BIK can be upregulated by the transcription factor E2F-1. Post-translationally, pro-apoptotic activity of human BIK is increased by phosphorylation [120]. Broad institute gene expression database (https://portals.broadinstitute.org/ccle/page?gene=BIK (accessed on 9 October 2020)) shows relatively high levels of BIK mRNA in prostate cancer cells, yet there are very few reports on the role of BIK in apoptosis regulation in prostate cancer cells [155,156], and no data on BIK protein levels in prostate tumors.

BID—BID plays a unique role among other BH3 only proteins by connecting an extrinsic death receptor activated apoptotic pathway with an intrinsic mitochondrial pathway.

The pro-apoptotic function of full-length, inactive BID is exposed by the post-translational cleavage of BID by a several proteases (such as caspase-8/-2 and granzyme B); thereby, forming an N-terminally truncated form of BID (tBID) that is targeted to the mitochondria. Post-cleavage N-myristylation of BID promotes its targeting to the mitochondrial outer membrane which then further enhances BID’s pro-apoptotic activity [101]. Upon targeting to the mitochondria, BID reportedly interacts and directly activates BAX and BAK [62,157] or according to more recent evidence binds anti-apoptotic BCLX and MCL1 and thus, shifts the balance toward apoptosis [158]. BID levels are regulated at the transcriptional level by the transcription factors TP53 [159] and HIF1 [160].

Immunohistochemical analysis of primary prostate cancers detected BID in normal prostate epithelium with higher immuno-scores then in advanced prostate cancers. In the T3 group higher BID scores were associated with longer recurrence-free survival. Analysis of the NCI panel of cancer cell lines showed lower BID levels in prostate cancer cells [161]. Several publications assessed death receptors signaling that was activated by FAS, TNF-α and TRAIL ligands in prostate cancer cells, and reported on the role of BID cleavage in defining an apoptosis decision [162,163,164,165,166]. The expression of C-terminal part of cleaved BID was sufficient to induce apoptosis in prostate cancer cells [164].

In summary, the evidence from the analysis of prostate cancer cell lines, transgenic mice and clinical samples support the role of the BCL-2 family proteins in apoptosis regulation in the prostate epithelium, yet our knowledge of specifics is limited. Experimental tools including inducible tissue-specific transgenes and single cell transcriptomics are now available to assess the role of BCL-2 family proteins in apoptosis that is induced by androgen ablation in the prostate epithelial cells. Several studies have looked into transcriptome changes in response to ADT [167,168,169] but none has reported changes in BCL-2 family gene expression. Perhaps, post-translational mechanisms are playing a leading role in regulating proteins of the BCL-2 family.

Experimental data from PCa cells provide compelling evidence on the regulation of MCL1, BAK and BH3-only proteins BAD and BIM by signal transduction pathways amenable for pharmacological targeting (Figure 2). The signal transduction pathways implicated in initiation and progression of the CRPC that target proteins of the BCL-2 family are described below.

## 4. Signaling Pathways That Can Control BCL-2 Family Proteins

### 4.1. PI3K/AKT Signaling Pathway

The PI3K/AKT is a conserved signal transduction pathway that controls cell growth, survival, angiogenesis, proliferation, metabolism, protein synthesis, and differentiation [170]. PI3K, a kinase belonging to the family of lipid kinases, is activated primarily by receptor tyrosine kinases (RTKs) at the cell membrane [11,138]. PI3K can also be activated by GPCRs, non-receptor tyrosine kinases and RAS oncogene [11]. Activated PI3K phosphorylates the 3′-hydroxyl group of phosphatidylinositol-4,5-diphosphate (PIP2); thereby, forming phosphatidylinositol-3,4,5-triphosphate (PIP3). PIP3 recruits the two protein kinases containing pleckstrin homology (PH) domain (serine/threonine kinases AKT and PDPK1/PDK1) to the membrane. AKT is then phosphorylated and activated by PDK1 and mTORC2 [11,138,171]. The phosphatase and tensin homolog deleted on chromosome 10 (PTEN) is a primary negative regulator of PI3K pathway that dephosphorylates PIP3 to PIP2, acting as a tumor suppressor [11,138].

The AKT phosphorylation creates binding sites for 14-3-3 chaperons that retain proteins in cytoplasm [172]. Direct targets of AKT include BAD and BIMEL [173,174] that are involved in mitochondrial apoptosis; and PACS2 that controls interorganelle protein traffic and BID cleavage downstream of death receptors [175]. AKT and related SGK kinase can also phosphorylate and prevent nuclear entry of FOXO transcription factors; thus, inhibiting expression of FOXO-regulated pro-apoptotic proteins BIM, PUMA, FASL and TRADD [176,177,178,179]. The upregulation of MCL1 downstream of PI3K by transcriptional [180,181] and translational [182] mechanisms have been also reported. AKT phosphorylation inhibits GSK3 kinase that creates binding sites for ubiquitin ligases and targets MCL1 to degradation [183]. Thus, PI3K signaling may increase MCL1 protein levels by controlling transcription, translation and degradation.

Upregulation of the PI3K signaling pathway has been observed in 30–50% of prostate cancers [4,11,12,138,184]. The inactivation of PTEN, is the most common alteration resulting in overactivation of the PI3K pathway. Inactive PTEN can result from deletions or more commonly from mutations in PTEN [138]. Mice with prostate-targeted PTEN knockout show increased PI3K/AKT activation and developed prostate cancer [185].The activation of PI3K pathway could also occur due to overexpression or activating mutations of AKT or PI3K [11].

In PCa cells active PI3K/AKT pathway, inhibited the mitochondrial apoptosis by phosphorylating BAD [130] and by inhibiting TRAIL-induced apoptosis upstream of BID cleavage by an undefined mechanism [162] (Figure 2).

### 4.2. RAS/ERK Signaling Pathway

The activation of the RTK/RAS/ERK pathway by a wide range of extracellular stimuli is understood in detail. The RAS/ERK pathway induces phosphorylation of downstream substrates by the extracellular signal-regulated kinases (ERKs) that control proliferation, survival, differentiation, angiogenesis, motility and invasiveness [186]. Active ERKs also phosphorylate and activate the RSK family kinases that in turn phosphorylate a variety of cytoplasmic and nuclear substrates including the BH3-only proteins BIM and BAD, the transcription factor CREB, and the vasodilator-stimulated phosphoprotein (VASP) that controls cytoskeletal assembly and cell survival (Figure 2) [187,188,189].

Although rarely found mutated, activation of the RAS/ERK signaling pathway correlates with an increased Gleason score, AR-independence, metastasis and poor prognosis [13,186,190,191]. Thus, PCa patients who did not respond to ADT were found to have high levels of phospho-ERK [190]. Over the past 20 years, data linking activation of ERK signaling to PCa progression and therapy resistance has been consistently reported [190,191,192,193]. Using *Nkx3.1;Pten* mutant mice, which resemble main characteristics of CRPC, it has been demonstrated that AKT and ERK signaling pathways are active in androgen independent lesions and act synergistically to promote CRPC [194]. In prostate cancer cells, the activation of RAS/ERK signaling inhibited apoptosis by phosphorylating BAD [126,131,137] and by suppressing BIM expression [148] (Figure 2). Recent publications connected activation of FGFR with RAS/ERK and an inhibitor of differentiation (ID1) transcription factor signaling that played a major role in CRPC progression independently from AR axis [17,195].

### 4.3. GPCR/PKA Signaling

In addition to PI3K/AKT and RAS/ERK signaling, the new evidence has emerged that connected signaling by GPCR/PKA module to prostate cancer progression and therapy resistance via inhibition of apoptosis in tumor epithelial cells and stem cells, and increased angiogenesis [21,196,197,198].

#### 4.3.1. VIP/PKA Signaling

The VIP is a neuropeptide widely distributed in the peripheral and central nervous systems as well as endocrine-paracrine cells [199,200]. Nerves supplying VIP are found in several organs including the prostate gland [200]. VIP binds to and activates VPACs receptors of the GPCRs family and PKA. VIP/PKA signaling protects the PCa cells from apoptosis by inducing BAD phosphorylation [130,196]. VIP was also shown to transactivate AR in a PKA-dependent manner [201].

#### 4.3.2. ADRB2/PKA Signaling 

β-adrenergic receptors (ADRBs), are seven transmembrane GPCRs activated by the catecholamines epinephrine (Epi) and norepinephrine (NE) released by the sympathetic nervous system (SNS) in response to physical and psychoemotional stress [197,198]. Epi is produced by chromaffin cells while NE is produced by adrenergic nerves that innervate most of the major organs including the prostate gland [198]. Activation of ADRB2 and downstream PKA by Epi and NE was found to be important for both normal prostate development and prostate carcinogenesis [197,198].

The prostate is a highly innervated organ with majority of the nerves found in the peripheral zone of the gland. In addition to that, the paraganglia, containing chromaffin cells that produce Epi, are located close to sympathetic nerves in the prostate. Secretions of luminal cells are facilitated by adrenergic nerve firing that activates the ADRBs [198]. ADRB2 is the predominant adrenergic receptor subtype in luminal cells of the human prostate while both ADRB2 and ADRB3 are found in prostate stromal cells [197,198].

The mechanisms of ADRB2 and VPAC GPCR signaling in PCa cell lines are similar. Ligand binding induces conformational change in GPCR that increases its GDP/GTP exchange activity toward the Gα subunit of heterotrimeric G-proteins associated with receptors. In GTP-bound form Gα binds to and activates adenylyl cyclase, that produce cAMP. cAMP binds to the regulatory subunit of PKA, causing the release of active PKA’s catalytic subunit. PKA’s catalytic subunit can then phosphorylate several downstream targets including BAD, MCL1, VASP and transcription factor CREB; and also modulate activation of PI3K/AKT and RAS/ERK pathways [21,198]. The ability of GPCR/PKA signaling to inactivate BAD and stabilize MCL1 results in more efficient inhibition of apoptosis compared to pathways that target just one molecule of the apoptotic machinery (Figure 2) [21].

Several recent studies demonstrated a connection between the activation of ADRB2/PKA pathway and therapy resistance in preclinical models of prostate cancer. Additionally, retrospective studies show that the use of beta-blockers is associated with the increased survival of patients with melanomas [202,203] and ovarian cancer [204], as well as prevention of metastasis and improvement of relapse-free survival of breast cancer patients [205]. In PCa the use of beta-blockers use has been linked with decreased metastasis, [206] decreased mortality in patients with high-risk or metastatic disease, [207] and improved survival in ADT-treated patients [208]. One such beta-blocker, propranolol, showed protective effect in several cancers including PCa [209]. Propranolol (a non-selective antagonist of ADRB1 and ADRB2) is clinically approved for the treatments of hypertension, cardiovascular diseases, anxiety and tremor [209,210] and could be repurposed for PCa treatment [21].

## 5. Clinical Trials of Drugs Targeting Anti-Apoptotic Signaling in PCa

In August 2020, 736 clinical trials of drugs targeting signaling pathways connected with the apoptosis inhibition in CRPC (RTK, RAF, and PI3K/AKT/MTOR) were registered in clinicaltrials.gov database. Numerous phase I studies assessed the dosage and safety of those drugs, yet we decided to focus on the results of 53 phase II/III trials published in peer-reviewed journals (Table 1). These clinical trials can be divided into four large groups: RTK inhibitors, VEGFR inhibitors, RAF kinase inhibitors and PI3K pathway inhibitors.

Activation of the PI3K/AKT/MTOR pathway is very common among CRPC patients, therefore, targeting components of this pathway could test the PI3K pathway contribution to therapy resistance in CRPC. A phase II study of the pan-AKT inhibitor ipatasertib showed a prolonged radiographic progression-free survival (rPFS) compared to a placebo particularly in PTEN-negative patients. The trend toward a longer overall survival and a time-to-PSA progression in the ipatasertib group was noted; yet, it did not reach a statistical significance [211]. Phase II trial of PI3K inhibitor buparlisib did not show prolongation of PFS when compared to historic data, and reported a 5-fold decline in plasma concentration of buparlisib when administered with enzalutamide [212]. Likewise, phase II trials with the pan-class I PI3K inhibitor PX-866 reported no resistance reversal when administered with abiraterone acetate in recurrent or mCRPC [213]. No phase III trials with anti-PI3K agents have been conducted so far, and published phase II trials did not include control groups, therefore, no conclusion on clinical benefits of targeting PI3K in CRPC can be made. Several clinical trials testing PI3K inhibitors are ongoing, including two phase I studies with AZD8186 (NCT03218826), (NCT01884285), and a phase II trial of dual inhibitor of class I PI3K and MTOR LY3023414 (NCT02407054), in advanced/metastatic PCa and CRPC.

Inhibitors of MTOR were tested in several phase II trials as single agents and in combination with other therapies for CRPC [214,215,216,217,218]. Comparison of pre-treatment and post-treatment biopsies in patients who received subtoxic dose of MTOR inhibitor MLN0128, detected a decrease in rpS6, but no change in pS473AKT or 4EBP1 [214]. None of the nine patients in this study showed a decrease in PSA or a circulated tumor cell (CTC) count. Lack of clinical activity and a transient CTC decline was reported in earlier phase II trials of MTOR inhibitor temsirolimus, yet phosphorylation of MTOR downstream targets was not evaluated in these studies [215,216,217]. Lack of clinical activity was also reported for everolimus [219], although the trend to a longer PFS was observed for patients with PTEN deletion [220]. Since activation of AR axis was connected with an inhibition of PI3K/MTOR signaling, a combination of MTOR inhibition with second generation AR inhibitors was suggested. Still, combining everolimus with bicalutamide proved ineffective in CRPC patients that had been previously treated with bicalutamide [221]. However, in the bicalutamide-naïve CRPC group, a higher percentage of patients showed a PSA decline compared to historical data for bicalutamide monotherapy [222]. Inhibiting MTOR (everolimus) together with EGFR (gefitinib) resulted in a rapid increase in PSA in 13/37 CRPC patients which then declined upon treatment discontinuation [223]. No evident clinical benefits were observed when MTOR inhibitor ridaforolimus was combined with taxane in CRPC patients [224]. In summary, phase II studies failed to demonstrate clinical efficacy of MTOR inhibitors that would justify phase III trials.

Overall, compared to PI3K and MTOR inhibitors, pan-AKT inhibitors demonstrated the best clinical efficacy against CRPC, particularly in PTEN-deficient tumors. These encouraging results highlight the importance of patient pre-screening to select patients with active PI3K pathway for clinical trials of inhibitors that target the PI3K pathway and also of the need to confirm the inhibition of the intended target by analyzing the relevant biomarkers in post-treatment biopsies.

Activation of receptor tyrosine kinases (RTK) was connected with CRPC, and several RTK inhibitors showed clinically meaningful activity in CRPC. The pan-RTK inhibitor cabozantinib (targeting MET, RET, AXL, VEGFR2, FLT3, and KIT) [225] demonstrated clinical activity in the CRPC phase II trials as indicated by the improved PFS (23.9 vs. 5.9 weeks [226] and 5.5 vs. 1.4 months [227]), pain relief, bone scans, analgesic use, measurable disease, and bone biomarkers when compared to a placebo [228]. However, phase III trials compared cabozantinib to prednisone or mitoxantrone-prednisone in heavily treated CRPC patients did not demonstrate significant improvements in OS or PSA outcomes despite improved PFS [229] or improve pain relief [230]. Unlike cabozantinib, the EGFR family inhibitor afatinib, did not result in any progression-free CRPC patients or PSA response in a phase II study [231].

Sunitinib (inhibitor of KIT, FLK1/KDR and PDGFR-beta RTKs) had minimal impact on PSA levels in a chemotherapy-naïve and docetaxel-treated CRPC patients, and had discordant PSA and radiographic responses. The target inhibition assessment revealed reductions in VEGFR2 and PDGFaa upon sunitinib treatment; however, PFS or OS was not assessed [232]. The EGFR inhibitor, gefitinib did not show objective responses in CRPC as a single agent or in combination with androgen-targeting therapy or docetaxel [233,234,235,236,237,238,239], while lapatinib, an EGFR and HER2 inhibitor, decreased blood PSA when compared with historic controls in CRPC patients [240]. In phase II studies of EGFR tyrosine kinase inhibitor erlotinib, PSA decreases were observed in CRPC patients, yet no improvements in OS or PFS were reported [241,242,243]. Additionally, phase II studies showed that combining the VEGFR-2/EGFR tyrosine kinase inhibitor vandetanib with bicalutamide or docetaxel/prednisolone did not have superior activity over bicalutamide or docetaxel/prednisolone monotherapy in mCRPC patients [244,245]. However, in metastatic CRPC, combining cetuximab (a monoclonal antibody directed against EGFR) with docetaxel increased PFS in patients that had overexpressed EGFR and persistent PTEN expression [246]. In another trial, cetuximab combined with doxorubicin caused minimal PSA declines, yet improved survival when compared to historic control groups [247]. Combining cetuximab with mitoxantrone/prednisone in unselected docetaxel-treated CRPC patients showed a shorter median time to progression comparing to patients who did not receive cetuximab (4.9 vs. 6.6 months); however, in cetuximab group in patients who developed a rash, time to progression was significantly longer (10.3 months vs. 2.8 months, *p* = 0.004) [248].

The HER2 dimerization inhibitor pertuzumab did not result in objective responses, but improved survival when compared to historic control groups [249]. Trastuzumab, a humanized monoclonal antibody directed against HER2, demonstrated poor efficacy [250,251], that could be due to low levels of HER2 expression in prostate cancer [252].

To sum up, based on results of trials that included control groups, broad specificity RTK inhibitors (e.g., cabozantinib) and EGFR-targeting monoclonal antibodies (e.g., cetuximab) showed the most clinically meaningful activity in CRPC patients by extending PFS based on bone scans and pain symptoms, yet failed to demonstrate a statistically significant increase in OS. Cetuximab (antibodies against EGFR) and trastuzumab (antibodies against HER2 that inhibit ligand-induced heterodimerization with other HER family members) showed a tendency to increased survival when compared to historic data.

Since neovascularization is required for growth of solid tumors [253], several VEGFR tyrosine kinase inhibitors have been tested in the clinic (Table 1). Phase II/III trials that included control groups showed that targeting angiogenesis by blocking all VEGF receptors (VEGFR-1, -2, -3) or VEGF-A improved PFS in CRPC, but failed to increase OS. Comparison with currently accepted protocols for CRPC treatments showed increased toxicity of VEGFR but no significant survival benefits [254].

Targeting RAF kinases has been tested in phase II trials in CRPC patients. In earlier trial, abrogation of RAF1 by the anti-sense oligonucleotide ISIS 5132 did not induce any objective or PSA responses in CRPC; although, tumor biopsies were not taken to evaluate the target knockdown [255]. A phase II trial of sorafenib (RAF and VEGF-2/PDGFR-beta kinase inhibitor) suggested clinical benefits for CRPC patients with metastatic bone lesions [256,257]. The analysis of bone marrow biopsies did not show a significant decrease of pERK (a downstream target of RAF kinases), but showed pERK reduction in one patient with improved bone scan lesions post-sorafenib treatment. However, biopsies were taken only after sorafenib treatments, thus, no comparisons in the same patient were possible [256]. Despite reported benefits and low toxicity in phase II trials, no phase III trials with RAF-targeting agents have been conducted to provide reliable information on the clinical efficacy of RAF kinase inhibitors in CRPC.

At present, there are no completed clinical trials targeting components of the RAS/ERK pathway (other than RAF kinases), the ADRB2/PKA pathway, or death receptor signaling. However, there are currently ongoing trials testing the efficacy of the MEK inhibitor trametinib in CRPC alone (NCT02881242) and with/without AR inhibition (NCT01990196). Clinical trials that examine the ADRB2 antagonists propranolol (NCT03152786) and carvedilol (NCT02944201) in primary PCa are ongoing; however, no clinical studies have been reported in CRPC patients.

In summary, at present the strongest effects on PFS are achieved with the broad specificity RTK inhibitor cabozantinib, EGFR-targeting monoclonal antibody cetuximab in PTEN positive tumors, and the pan-AKT inhibitor ipatasertib. Importantly, more pronounced improvements with AKT and MTOR inhibitors were observed in patients with PTEN-negative tumors (where PI3K pathway is constitutively active). Effects on overall survival were modest or not yet available. Although several phase II studies tested PI3K, MTOR and RAF-1 inhibitors, none of those studies provided reliable information on clinical efficacy of those drugs and future studies with control groups are needed to prove or disprove their efficacy in CRPC.

These rather disappointing results are a call to rethink the way clinical trials of signal transduction inhibitors are designed. Patient selection based on tumor molecular profiling; monitoring inhibition of target molecules and downstream pathways along with changes in signaling network topology, apoptosis and proliferation before and during treatment are just a few examples of the “to do list” for future clinical trials. New tools for transcriptome, proteome and phospho-proteome analysis at the single cell level are needed to provide meaningful data for the clinical evaluation of inhibitors of signaling pathways that control apoptosis in CRPC.

**Table 1 cancers-13-00937-t001:** Clinical trials of drugs targeting anti-apoptotic signaling in prostate cancer (PCa).

Agent	Target(s)	Population	Phase	Outcome
Ipatasertib (GDC-0068, RG-7440) + abiraterone	All AKT isoforms + AR	PTEN-negative mCRPC	Phase II	• Improved rPFS (8.18 mo (400 mg ipatasertib) and 8.31 mo (200 mg ipatasertib) vs. 6.37 mo for placebo) [212]
MLN0128 (Sapanisertib, INK 128)	TORC1 and TORC2	mCRPC—after abiraterone acetate and/or enzalutamide	Phase II	• PSA rise on treatment in all patients (median, 159% increase from baseline); PSA declined immediately in 4 patients upon drug discontinuation [215]
Temsirolimus (Torisel, CCI-779)	MTOR	mCRPC	Phase II	≥30% PSA decline (1 patient). Median PFS: 1.9 mo; median OS: 8.8 mo [216]
Everolimus (RAD001, Afinitor, Zortress)	MTOR	chemotherapy-naïve mCRPC patients	Phase II	≥50% PSA response (5% of patients), ≥30% PSA decline (11% patients) Improved PFS and response associated with PTEN deletion [221]
Everolimus + bicalutamide	MTOR +AR	CRPC—previously treated with bicalutamide	Phase II	2/36 patients had a confirmed ≥50% PSA reduction; 9/36 patients had stable disease (SD) by PSA levels (no longer than 6 mo) Did not reach primary endpoint of improved response compared to historic bicalutamide results [222]
		Bicalutamide-naïve CRPC	Phase II	• ≥30% PSA decline in 75% of patients [223].
Everolimus + carboplatin + prednisone	MTOR + DNA replication + GR	mCRPC pretreated with docetaxel	Phase II	Median time to progression: 2.5 mo; Median OS: 12.5 mo 8/10 patients with positive nuclear phosphorylated pAKT progressed within 9 weeks while 2/10 patients with negative staining continued without progression for 30 and 48 weeks [220]
Buparlisib (BKM-120) + enzalutamide	Class I PI3K + AR	mCRPC progressing on or are not candidates for docetaxel	Phase II	6 mo median PFS: 1.9 mo (buparlisib alone) vs. 3.5 mo (buparlisib/enzalutamide) Median OS of all subjects: 10.6 mo (similar buparlisib or buparlisib + enzalutamide groups). 1-year survival rate: 35.3% (buparlisib) and 53.8% (buparlisib/enzalutamide) PSA declined in 23% of patients (not ≥50%); decline rates did not differ in with/without enzalutamide No radiographic responses [213]
PX-866 +/− abiraterone acetate	Class I PI3K + AR	recurrent or metastatic CRPC	Phase II	With single agent PX-866, 33% were progression-free at 12 weeks, 2 partial OR, and 1 confirmed PSA response Combination showed no evidence of castration resistance reversal (24% patients were progression-free at 12 weeks; no OR or PSA response). No correlation between PTEN status and response [214]
Temsirolimus (Torisel, CCI-779)	MTOR	Docetaxel-treated CRPC patients	Phase II	Median time to treatment failure: 24.3 weeks; Median time to PSA progression: 12.2 weeks 1 partial tumor response (4.8%); 1 PSA response (4.8%) [217].
		Chemotherapy-naïve CRPC		Partial response (PR): 2/15 patients; SD: 8/15; OR: 2/15; Overall clinical benefit (OR+SD): 10/15. Median time to radiographic disease progression: 2 mo Any PSA decline: 4/14 four patients (28.5%); 1 patient (7%) had >50% PSA decline. Median time to PSA progression: 2 mo Median OS: 13 mo; 3 patients remained alive at the data cutoff for an OS of 14% at 4 years [218].
Ridaforolimus (AP23573 MK8669)	MTOR	Taxane-treated CRPC patients	Phase II	No OR; 47.4% of patients had SD as their best response; 21.1% had SD as their best overall PSA response. Median time to progression: 28 days [225]
Ridaforolimus + bicalutamide	MTOR + AR	AsymptomaticmCRPC patients	Phase II	Dose reductions were required in 7 patients. 3/11 patients experienced dose-limited toxicity leading to <75% of planned ridaforolimus dose during the first 35 days of study treatment. The pharmacokinetic results showed no differences in exposures to ridaforolimus with and without concomitant bicalutamide administration [219].
Everolimus + gefitinib (Iressa, ZD 1839)	MTOR + EGFR	CRPC	Phase I/II	67% discontinued treatment before evaluation due to progression (evidenced by PSA levels or imaging or because of a grade 2 or higher toxicity). 35% rapidly elevated PSA which declined upon treatment discontinuation. Fluorodeoxyglucose positron emission tomography 24 to 72 h after the initiation of treatment showed a decrease in the standardized uptake value consistent with MTOR inhibition in 27 of 33 evaluable patients (82%); there was a corresponding rise in PSA in 20 of these 27 patients (74%) [224].
cabozantinib (Cabometyx, Cometriq, Cabozanix, BMS-907351, XL184)	RTK	advanced, recurrent or metastatic cancers	Phase II	• CRPC patients had largest PFS (median, 5.5 mo vs. 1.4 mo for placebo) [228]
		CRPC	Phase II	• Improved PFS (median, 23.9 weeks with cabozantinib vs. 5.9 weeks with placebo) [227].
				clinically meaningful pain relief: 57% of patients; reduction or discontinuation of narcotic analgesics: 55%; bone scan response improvement: 73% (100 mg) and 45% (40 mg); reductions in measurable soft tissue disease: 80% and 79%, respectively. Median OS: 12.1 mo (100-mg) and 9.1 mo (40-mg) [229].
		mCRPC—previously treated with docetaxel and abiraterone and/or enzalutamide	Phase III	Improved OS (11.0 mo with cabozantinib vs. 9.8 mo with prednisone) and rPFS (median, 5.6 v 2.8 mo) compared to prednisone No improvement in PSA outcomes compared to prednisone [230]
Sunitinib (Sutent, SU11248, SU011248)	RTK	CRPC	Phase II	1 confirmed 50% PSA decline in chemotherapy-naïve group (A) and one in docetaxel-resistant CRPC groups (B). 8 and 12 patients in groups A and B, respectively had stable PSA upon treatment. Improvements in imaging were seen post-treatment despite lack of PSA declines [233].
Afatinib (Gilotrif, BIBW 2992 MA2)	EGFR	CRPC	Phase II	• limited anti-tumor activity [232]
Gefitinib (ZD1839, Iressa)	EGFR	mCRPC progressing on LHRH analog + antiandrogen (bicalutamide or flutamide).	Phase II	• No PSA or objective response in any patient [236].
		nmCRPC	Phase II	• None of the evaluable patients had a PSA response [238].
		Chemotherapy-naïve CRPC	Phase II	No patient had PSA or objective measurable response; 14.3% of patients had stable PSA (1 patient at 250 mg gefitinib and 4 at 500 mg); 14.3% had a best response of SD (duration, 2.5 to 16.8 mo). No significant effect on the rate of PSA increase [234].
		CRPC	Phase II	1 patient had a confirmed 50% PSA response. Median time to progression: 28 days. Majority of patients had a stable performance status while on study. 13/51 patients required a dose reduction; 9/51 patients withdrew due to an adverse event [259].
Gefitinib + docetaxel	EGFR + tubulin	CRPC	Phase II	• Response rate and duration were consistent with those of docetaxel monotherapy [237]
Gefitinib + prednisone	EGFR + GR	CRPC	Phase II	At median follow-up time of 29 mo, 77 patients progressed and 51 died. PSA response in 15.8% (combination) and 11.4% patients (prednisone + placebo) Time to progression: median, 4.0 mo (prednisone + gefitinib); 4.5 mo (prednisone + placebo); survival: median, 26.5 mo (prednisone + gefitinib); 20.5 mo (prednisone + placebo) [240].
Gefitinib + antiandrogen + LH-RH analogue	EGFR + AR + GnRH receptor	CRPC	Phase II	• Gefitinib treatment did not result in any objective response (PSA or OR). Median time to progression was 70 days. Median OS was 293 days [235].
Vandetanib (ZACTIMA) + bicalutamide	VEGFR2/EGFR + AR	Chemotherapy-naïve mCRPC (rising PSA on ADT and minimally symptomatic)	Phase II	PSA response: 18% in arm A (vandetanib + bicalutamide) and 19% arm B (bicalutamide). Time to PSA progression: 3.16 mo (Arm A) and 3.09 mo (Arm B) [245].
Vandetanib	VEGFR2/EGFR	mCRPC	Phase II	PSA response: 67% patients (placebo + DP) vs. 40% (vandetanib + DP) 65% experienced progression events (disease progression or death from any cause) in vandetanib + DP vs. 60% in placebo + DP [246].
cetuximab	EGFR	mCRPC	Phase Ib/IIa	SD in 65% patients with bone disease and 61% patients with lymph node disease PSA declines: modest in the 36 patients; 1 patient (2.7%) had an 80% decline from baseline, 2 (5.6%) had >50% to <80% decline Median survival: 18 mo [248].
cetuximab + mitoxantrone + prednisone	EGFR + topoisomerase II + GR	mCRPC progressing after docetaxel	Phase II	• Median time to progression (TTP): 4.9 mo (cetuximab + mitoxantrone + prednisone) and 6.6 mo (mitoxantrone + prednisone); measurable disease response rate: 2% and 4%; PSA response rate: 7.7% and 17.6%; median survival: 11.9 and 15.7 mo, respectively [249].
cetuximab + docetaxel	EGFR+tubulin	mCRPC	Phase II	7 patients (20%) and 11 patients (31%) had confirmed ≥50% and ≥30% PSA declines, respectively. Patients overexpressing EGFR and having persistent PTEN activity had significantly improved PFS [247].
Pertuzumab	HER2 dimerization	CRPC progressing after at least one taxane-based regimen	Phase II	No complete or partial responses (as defined by Response Evaluation Criteria in Solid Tumors Group or 50% decline in PSA). 5/30 patients had SD for at least 23 weeks; 1/5 had SD for 36 weeks. Survival was prolonged with pertuzumab (16.4 mo), compared with historic controls with similar baseline prognostic features (10.7 mo) [250].
Lapatinib (Tykerb, GSK572016, GW2016, GW-572016)	EGFR and HER2	Chemotherapy-naïve CRPC patients with rising PSA on ADT	Phase II	1/21 patients had >50% PSA decline; another patient had 47% PSA decline (45+ mo) Median time to PSA progression: 29 days [241].
Erlotinib (Tarceva, CP-358,774, CP358774, OSI774, OSI-774)	EGFR	Advanced or metastatic PCa (including CRPC)	Phase II	No patients had a PSA decline, 14% had stabilized PSA. PSA-doubling time increased in 10 patients Clinical benefit was achieved in 40% of patients [244].
		chemotherapy-naive CRPC	Phase II	2/22 patients had PR, 5 had SD; overall clinical benefit: 31%. PSA-doubling time: increased from 3 mo (before study) to 6 mo in all responding patients. 2 patients had >50% PSA decline Median time to disease progression: 2 mo; at 12 mo, 9% of patients were progression-free. Median OS: 16.3 mo; 1- and 2-year survival rates: 58% and 27%, respectively [243].
Erlotinib + docetaxel	EGFR + tubulin	≥ 65 years CRPC patients progressing despite ADT	Phase II	No OR in 8 patients with measurable lesions. 1 patient (5%) had bone scan improvement and PSA decline; 5/22 patients experienced ≥50% PSA decline [242].
Trastuzumab (Herceptin)	HER2	CRPC	Phase II	2 patients had SD (based on PSA decline to <50% of baseline). No patient demonstrated a regression of radiographic bony or soft tissue metastatic disease [251].
Trastuzumab + docetaxel	HER2 + tubulin	HER2-positive CRPC		No patient responded to trastuzumab alone. Median survival was not reached; median PFS was 7 mo. Trial was closed for non-feasibility [253]
Cediranib	VEGFR	mCRPC—progressing following docetaxel therapy	Phase II	• 6/39 patients had confirmed PR; 1 had an unconfirmed PR. 43.9% of patients were progression-free at 6 mo; for all patients, median PFS: 3.7 mo; OS: 10.1 mo [260]
Cediranib + DP	VEGFR	mCRPC	Phase II	• 6-mo PFS rate: 61% in DP+C arm and 57% in DP arm. There were no significant differences in 6-mo OS rate, objective tumor and PSA response rates, and biomarkers in the two arms [261].
Bevacizumab + temsirolimus	VEGF-A & MTOR	mCRPC—previously treated with chemotherapy	Phase I-II	• Median time to progression: 2.6 mo; median best PSA change: 32% increase (met the predefined futility rule leading to early termination of the study) [262]
Bevacizumab + DP	VEGF-A	mCRPC	Phase III	• Improved PFS (median, 9.9 mo for DP+B vs. 7.5 mo for DP) and OR (49.4% vs. 35.5%;) but not OS (median, 22.6 mo vs. 21.5 mo) [255].
bevacizumab + docetaxel + estramustine	VEGF-A + tubulin + MAPs and tubulin	CRPC	Phase II	75% of patients (58/77) had a 50% PSA decline; 23/39 patients (59%) had a PR. Median PFS: 8 mo; median OS: 24 mo [263].
Dovitinib (CHIR-258, TKI258)	VEGFR, FGFR, PDGFR	mCRPC (80% were post-docetaxel)	Phase II	Median PFS: 3.67 mo; median OS: 13.70 mo. Chemotherapy-naïve patients had longer PFS (17.90 mo) compared with docetaxel-treated patients (2.07 mo) Patients with high serum VEGFR2 level over median level had longer PFS (6.03 mo vs. 1.97 mo) [264].
Pazopanib (Votrient, GW786034B) + bicalutamide	VEGFR + AR	chemotherapy-naive (CRPC).	Phase II	In arm A (pazopanib), 1 patient (11%) had a PSA response (confirmed ≥50% decline from baseline), 3 (33%) had stable PSA, and 5 (56%) had PSA progression; In arm B (pazopanib + bicalutamide), 2 patients (17%) had PSA responses, 6 (50%) had stable PSA, and 4 (33%) had PSA progression. Median PFS: 7.3 mo (similar in both arms); Long-term SD was seen in 4 patients remaining on treatment for 18 mo and 26 mo (arm A) and 35 mo and 52 mo (arm B) [265].
Aflibercept (Eylea, Zaltrap) + Docetaxel + prednisone	VEGFA/B & PGF + tubulin + GR	Chemotherapy-naïve mCRPC	Phase III	• Median OS: 22·1 mo (aflibercept + DP) and 21·2 mo (DP) no clinical benefit compared to control group [266].
SU5416 + dexamethasone	VEGFR2	Chemotherapy-naïve CRPC	Phase II	• No detectable effect on PSA secretion or time to progression [267].
Thalidomide (Contergan, Thalomid, Talidex)	pro-angiogenic factors (e.g., VEGF, BFGF, IL-6	mCRPC (failed other therapy forms)	Phase II	• ≥50% PSA decline in 18% of patients (200 mg/day) but no decline in the high-dose arm (200 mg/day to 1200 mg/day); PSA decline was maintained for >150 days in 4 patients; 28% of all patients had >40% PSA decline; 13% had 40% to 50% PSA decline [268].
				3 men (15%) had minimum 50% PSA decline (sustained throughout treatment); 6/16 (37.5%) patients (treated for at least 2 mo), had a median PSA decline of 48%. Increasing levels of serum bFGF and VEGF were associated with progressive disease (5/6 men with PSA decline also demonstrated decline in bFGF and VEGF levels and 3/4 men with a rising PSA showed an increase in both growth factors [269].
Itraconazole	Angiogenesis Hedgehog signaling	chemotherapy-naïve mCRPC	Phase II	• PSA PFS rates at 24 weeks: 11.8% in 200 mg/day arm and 48.0% in the 600 mg/day arm; median PFS: 11.9 weeks and 35.9 weeks, PSA response rates: 0% and 14.3%, respectively [270].
Sorafenib	RAF VEGFR2 PDGFRB	mCRPC	Phase II	13/21 patients with progressive disease progressed only by PSA criteria without clinical and radiographic progression. 2 patients met PSA progression criteria at the time when scans were obtained yet had dramatic reduction of bone metastatic lesions (shown by bone scan) [257]
				1/24 patients (21 previously treated with docetaxel) had a PR; 10 had SD. Median PFS: 3.7 mo; median OS: 18 mo (median potential follow-up of 27.2 mo:); median survival for the whole trial of 46 patients: 18.3 mo [258].
ISIS 5132	RAF1	chemotherapy-naïve, mCRPC	Phase II	No objective or PSA responses in any treated patient 1 patient had persistent SD 6 mo 2/16 patients did not have 25% or more rise in PSA 120 days post-therapy [256].

## 6. Conclusions

Numerous clinical trials have tested inhibitors of PI3K and ERK pathways in CRPC; however, the results of these trials were disappointing. Only the pan-RTK inhibitor cabozantinib, and VEGFR inhibitors aflibercept and bevacizumab, have advanced to stage III trials, yet these trials did not show significant clinical benefits. Perhaps the most telling lesson of these failures, is that the design of clinical trials should be guided by tumor analysis data. Several considerations for future clinical trials can be proposed.

First, patients should be pre-selected based on the evidence of activation of the signaling pathway for a clinical trial that test the inhibitor of this pathway. This approach is supported by the recent results on AKT inhibitors that showed clinical benefits only in patients with active PI3K pathway [211]. Second, the efficacy of the inhibition of the intended target should be monitored. This requirement is more difficult to implement in clinical setting as it requires repeated tumor biopsies or reliable surrogate biomarkers (for example, analysis of hair follicles biopsies for EGFR phosphorylation and GLI levels to control pharmacodynamics of EGFR and Hedgehog inhibitors) [269,270]. Third, the analysis of the biologically relevant targets of signaling pathways should be considered. This is the most difficult condition to satisfy, since despite extensive tissue culture studies, in vivo validation of target proteins responsible for the effects of specific signaling pathways on prostate epithelial cell survival are lacking. Yet, without understanding of what these targets are, what signaling pathways control them, and what alternative signals may compensate for inhibition of the pathway tested in clinical trial, we cannot interpret the results of successful or of failed trials alike.

Recent data from single cell analysis of prostate epithelial and of prostate stroma cells in mice subjected to androgen ablation, provide convincing support for the indirect regulation of apoptosis in prostate epithelial cells through paracrine factors secreted by stroma cells with active AR signaling [35,45]. This paradigm shift highlights the importance of identifying signaling pathways that are activated by stromal paracrine factors in prostate epithelial cells and the apoptosis regulatory proteins targeted by these pathways. Uncovering how apoptosis is regulated in normal prostates, will guide the experiments that examine whether the same key proteins are regulated in CRPC via AR-independent mechanisms, and whether targeting these mechanisms will provide effective therapies for CRPC.

In prostate cancer cell lines, the proteins of BCL-2 family MCL1, BAK, BAD and BIM can be regulated by signal transduction pathways amenable for pharmacological targeting. Future experiments in transgenic mice with inducible knockout of these proteins, or knock-in of phosphorylation -deficient mutants can assess whether these proteins are responsible for the survival effects of paracrine factors that are regulated by androgens. Confirming the role of these BCL-2 family proteins in maintaining prostate homeostasis *in vivo*, will justify the clinical testing of these proteins as biomarkers of tumor response to experimental therapies that inhibit signaling pathways that control these proteins. *In vivo* studies of these critical apoptosis regulators are limited by lack of the reliable technologies that allow to measure protein expression and post-translational modifications at the single cell level. Although the potential of advanced proteomics for analysis of prostate clinical samples is recognized [269,271], the implementation of these technologies lags behind genomic technologies.

In conclusion, extensive experimental data point at the limited number of BCL-2 family proteins as possible targets of signaling pathways that regulate apoptosis in prostate epithelium. The analysis of these proteins in mouse models of prostate cancer, and in patients’ PCa samples, will clarify their role in CRPC pathogenesis; hence, forming a foundation for “smart” clinical trials of selective inhibitors that are guided by pathway-specific biomarkers. These clinical trials will bring about improved personalized therapies that extend the lives of PCa patients; or will show that the research focus should be directed on apoptosis-independent mechanisms of CRPC.

## Figures and Tables

**Figure 1 cancers-13-00937-f001:**
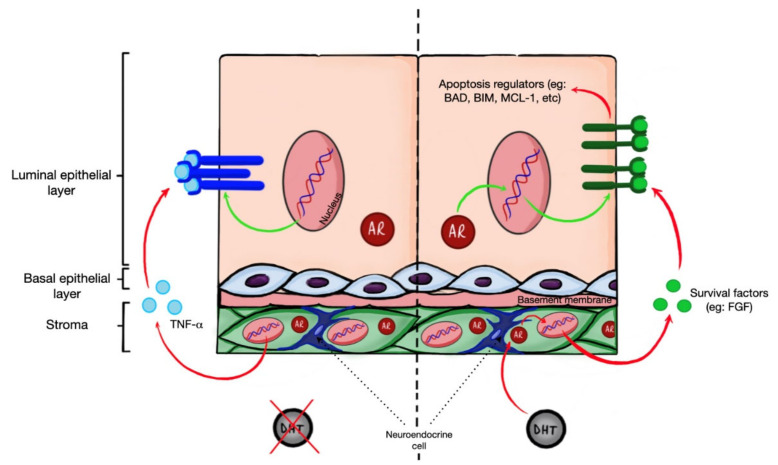
Androgen receptor (AR)-induced paracrine factors regulate apoptosis in prostate epithelial cells. Paracrine factors secreted by fibroblasts, neuroendocrine and other stroma cells regulate prostate tissue homeostasis. In stromal fibroblasts AR induces expression of survival factors that act on luminal epithelial cells. In prostate epithelial cells AR signaling increases expression of receptors of survival factors. Inhibition of AR signaling leads to increased production of TNF-α by stroma.

**Figure 2 cancers-13-00937-f002:**
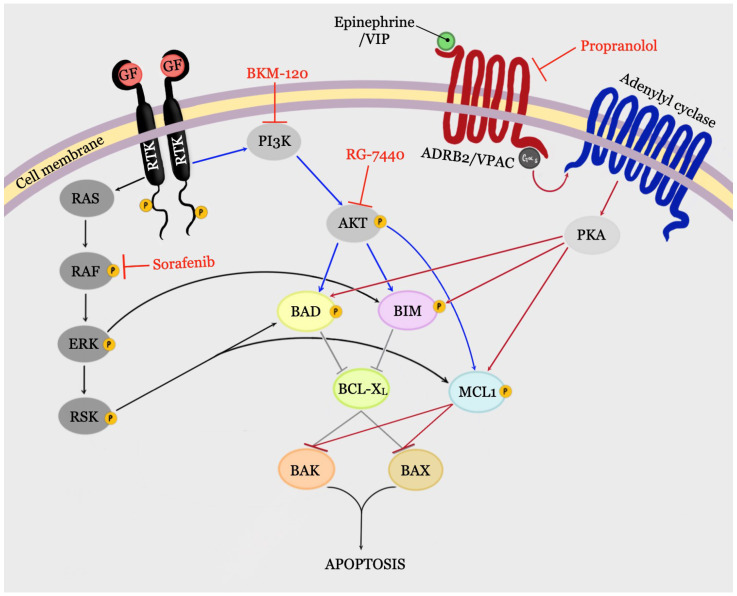
Apoptosis regulatory network in prostate cancer. MCL1 and BCLXL are dominant anti-apoptotic BCL2-family proteins in prostate cancer cells. RTK/ERK, PI3K/AKT and ADRB2/PKA signaling regulates transcription, translation and degradation of the anti-apoptotic protein MCL1 and pro-apoptotic functions of BIM and BAD. Inhibitors of these pathways are being tested in clinical trials.

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
