# Peer review of "Signaling Pathways That Control Apoptosis in Prostate Cancer"

_cancers, 2021, doi:10.3390/cancers13050937_

Round 1

Reviewer 1 Report

Authors have done a good work revising the manuscript according to the previous reviews and in the present form, the manuscript is certainly much clearer and complete and thus can be recommended for publication.

Reviewer 2 Report

This has been a new submission following a previous rejection. 

Tha authors have revised the manuscript accordingly and have conducted changes that are now compatible with accepting the manuscript.

This manuscript is a resubmission of an earlier submission. The following is a list of the peer review reports and author responses from that submission.

Round 1

Reviewer 1 Report

The review presented here is well-written and presented in great detail. The authors have presented a very detailed background on prostate cancer, including the clinical grades, histology, risk factors, and a review on recent developments in the signaling pathways in PCa apoptosis. The review is very significant and falls within the scope of the journal.

The initial background seems to be too detailed on basics and takes the focus away of the main topic, which is 'apoptotic signaling in PCa'. In fact, discussion on the main topic does not start until the 7th page of the review. So the initial discussion on histology, risk factors, etc can be shortened to give a concise background (most of the background material presented here is well-known already and several reviews already focused on it) followed by focused discussion on apoptotic signaling. This can improve the readability of the manuscript and will stick to the main topic.

Reviewer 2 Report

The review entitled “Signaling Pathways that Control Apoptosis in Prostate Cancer” attempts to cover a very large topic related to molecular mechanisms of prostate cancer development. Authors should put more effort into the Introduction section. This section does not introduce enough to the topic, does not explain why the authors dealt with it. Furthermore, too much of the publication has been devoted to topics other than the main topic in the publication title. I think that these topics should be more related to the process of apoptosis. I think the part describing propranolol is superfluous, I do not fully understand the legitimacy of adding this description to the manuscript.

Reviewer 3 Report

This work is an interesting and well organized reflection on the genetics, biochemistry and clinical findings of PC.

However, in my opinion a review should go beyond just a compendium of information and provide new insights onto the current state of the art, future directions and emerging technologies for diagnosis and treatment. While this work provides an interesting and well-organized reflection on the pathophysiology, genetics and biochemistry of PC, a significant number of previously-published reviews do the same. When reading the work, it is not clear how this works departs from others and what new insights and interpretations are provided. For this reason, i believe it cannot be accepted at this moment.

I leave below some suggestions to the authors:

Line 34-36 “In the kingdom of Saudi Arabia, PCa is the sixth most commonly 35 diagnosed malignancy among men of all ages and the most prevalent cancer in men over 75 years of 36 age [4,5].”

While it is understandable the interest the authors have for the specific case of Saudi Arabia, if this review intends to have a widespread view on the topic, it should not focus on a single country. Unless there is a specific reason to refer Saudi Arabia but not other countries, clearly state so.

Figure 2: Refer the staining used in A-C

Figure 3: Please provide the original images in high-resolution as supplementary data, as the interpretation of Figure 3 is difficult due to the reduced size of the panels.

Be sure to include the full meaning for all abbreviations. For example, TMPRSS2-ERG (line 178), PTEN (line 194) is not referred anywhere in the text. While genes do not require this, it would be useful for transcription factors such as ERG. This is not homogeneous throughout the paper. For example, lines 318-336 clearly state the meaning of all abbreviations.

What is the genetic basis of the influence of ethnicity in PC development? The authors refer the increased prevalence in black men and lower expression of TMPRSS1:ERG fusion gene. Is this information available?

It seems odd to have “Propranolol” as a stand-alone section. Perhaps this should be a general section in the likes of “Pharmacotherapy”? Are there no other examples? Surely propranolol is not the only drug with the beneficial effect in PC.